# IRF4 Mediates Immune Evasion to Facilitate EBV Transformation

**DOI:** 10.3390/v17070885

**Published:** 2025-06-24

**Authors:** Ling Wang, Culton R. Hensley, Jahan Rifat, Adam D. Walker, Katharine Ning, Jonathan P. Moorman, Zhi Q. Yao, Shunbin Ning

**Affiliations:** 1Department of Internal Medicine, Quillen College of Medicine, East Tennessee State University, Johnson City, TN 37614, USAmoorman@etsu.edu (J.P.M.); yao@etsu.edu (Z.Q.Y.); 2Center of Excellence for Inflammation, Infectious Diseases and Immunity, Quillen College of Medicine, East Tennessee State University, Johnson City, TN 37614, USA; 3Hepatitis (HCV/HIV) Program, James H. Quillen VA Medical Center, Johnson City, TN 37614, USA

**Keywords:** IRF4, EBV, PD1, PD-L1

## Abstract

The lymphocyte-specific transcription factor interferon regulatory factor 4 (IRF4) is a key player in immune evasion in cancers, with the complex mechanism(s) being barely understood. In this study, we have focused on the role of IRF4 in regulating T cell functions through its transcriptional regulation of programmed death 1 (PD1) and its ligand PD1 ligand 1 (PD-L1), which were identified as IRF4 transcriptional targets in multi-omics analysis. We have shown that IRF4 transcriptionally regulates both PD1 and PD-L1, promoting immune suppression in the context of Epstein–Barr virus (EBV) infection. Co-culturing EBV+ JiJoye lymphoma cells with CD4+ T cells or with peripheral blood mononuclear cells (PBMCs) downregulates CD4+ T cell functions, but the depletion of IRF4 in EBV+ JiJoye lymphoma cells reduces PD1 and PD-L1 expression, and partially restores CD4+ T cell functions. Moreover, CD4+ T cell depletion from PBMCs enhances EBV transformation, and EBV has a greater efficiency in transforming PBMCs from HIV patients with impaired CD4+ T cell functions. These findings support the role of IRF4 in immune evasion by upregulating PD1/PD-L1 during EBV transformation, and that functional CD4+ T cells are essential for limiting EBV transformation.

## 1. Introduction

The immune system plays a determinant role in carcinogenesis [1,2], including Epstein–Barr virus (EBV)-associated cancers [3,4,5,6,7]. As the most potent human tumor virus, EBV markedly increases the risk of various lymphomas in infected individuals with immune deficiencies, such as those caused by acquired immunodeficiency syndrome (AIDS) [3,8,9,10,11,12,13].

The interferon regulatory factor (IRF) family of transcription factors plays pivotal roles in the regulation of multiple facets of immune defense against pathogen infections and cancers [14]. The family member IRF4, known as multiple myeloma oncogene 1 (MUM1), is a potent, lymphocyte-specific oncogene, frequently mutated and aberrantly expressed in certain blood malignancies with lineage and stage specificities [15,16,17,18,19,20]. IRF4 overexpression is a hallmark of the activated B cell (ABC) type of diffuse large B cell lymphoma (DLBCL) [17,21]. In clinical practice, IRF4 serves as an important prognostic and diagnostic marker for these malignancies [16,18,19,22].

Chronic viral infection can result in an exhaustion state of T cells (Texh) expressing higher levels of immune inhibitory receptors (e.g., PD1 and LAG3) in the tumor microenvironment (TME) [23,24,25,26,27,28,29,30]. Texh cells, among other T cell subsets, play a central role in cancer immunity and in response to immune checkpoint blockade (ICB) therapies [31,32]. EBV-specific antigens are expressed in related malignancies and can be specifically targeted by T cells [33,34]. In this regard, EBV-specific T cells have produced exciting outcomes in treating post-transplant lymphoproliferative disorder (PTLD) in the last two decades, and many clinical trials have been launched in treating other EBV-related malignancies [25,35,36,37,38,39,40]. As such, PD1 and cytotoxic T lymphocyte antigen 4 (CTLA4) blockage successfully inhibited EBV-induced lymphoma growth in a humanized mouse model [41]. Thus, immunotherapy and certain treatments that boost the immune system can be more effective in managing virus-associated cancers [42,43,44,45,46,47,48]. Moreover, the ability of EBV to express T cell-specific antigens implies that EBV itself has potential applications in cancer immunotherapy, underscored by the evidence that ectopically expressed EBV latent membrane protein 1 (LMP1) induces multiple cellular tumor-associated antigens (TAAs) that trigger a robust CD4/CD8 cytotoxic response in EBV-unrelated cancers [49,50].

Accumulating evidence indicates that IRF4 governs anti-tumor T cell immunity [51,52,53,54]. IRF4-deficient mice failed to mount antibody, cytotoxic, and anti-tumor responses [55]. Recent efforts toward improving the efficacy and sustainability of anti-tumor immunotherapies include targeting the TME, which involves multiple transcription regulatory factors, including IRF4 [52]. The IRF4 antisense oligo INO251, in a clinical trial, prevented myeloma regeneration in mice [56]. Notably, IRF4 is expressed in 100% of LMP1-driven B lymphomas in mice [57]. Moreover, IRF4 is essential for EBV transformation of human B lymphocytes in vitro [58], and its depletion renders EBV-infected cells susceptible to apoptosis [59,60].

In recent years, using multi-omics approaches, we have published a series of original findings elucidating the mechanisms underlying the interaction between IRF4 and latent infections of the oncogenic viruses EBV and human T cell leukemia virus 1 (HTLV-1). Our studies demonstrated that LMP1 signaling induces the expression, site-specific phosphorylation, and activation of IRF4, which, in turn, upregulates a distinct set of target genes, including the B cell integration cluster (BIC), which encodes microRNA miR-155, and LIM domain-containing 1 (LIMD1) [59,60,61,62,63,64].

In this study, we showed that IRF4 transcriptionally induces both PD-L1 and PD1 expression in EBV lymphoma cells and co-cultured T cells, respectively, resulting in the evasion of the PD-L1/PD1 anti-tumor T cell response. Our findings unveil the role of IRF4 in immune evasion in viral oncogenesis.

## 2. Materials and Methods

### 2.1. Study Subjects

Twenty-seven whole-blood samples from local healthy subjects (HS, seronegative for HBV, HCV, and HIV) were obtained from BioIVT (Gray, TN, USA). The patient subjects for this study included twenty-three people living with HIV (PLHIV) on antiviral treatment (ART ) with undetectable viremia (HIV-RNA < 20 copies/mL of whole-blood), consisting of twelve HIV immune response patients (HIV-IR; >450 CD4+ T cells/µL of whole-blood) and eleven HIV immune non-response patients (HIV-INR; <450 CD4+ T cells/µL of whole-blood). Subjects with malignancy, transplantation, HBV or HCV infection, or immunosuppressive drug treatment were excluded. The information about the study subjects is shown in Table 1. 

### 2.2. Cell Isolation and Culture

Peripheral blood mononuclear cells (PBMCs) were isolated from whole blood by Lymphoprep^TM^ with SepMate^TM^-50 tubes (StemCell Technologies; Cambridge, MA, USA). CD4+ T cells were isolated from PBMCs using a CD4 T Cell Isolation Kit (Miltenyi Biotec; San Diego, USA); PBMCs/CD4- were isolated from PBMCs using CD4 microbeads (Miltenyi Biotec, San Diego, USA). BJAB is an EBV-negative Burkitt’s lymphoma (BL) cell line, and JiJoye is an EBV-infected BL cell line. Tet-inducible JiJoye-shControl (shCtrl) and JiJoye-shIRF4 cells were previously established [60]. The cells were cultured in RPMI-1640 medium containing 10% fetal bovine serum (FBS) (Gibco, Brooklyn, NY, USA), 100 IU/mL penicillin, and 2 mM L-glutamine (ThermoFisher Scientific, Logan, UT, USA).

### 2.3. EBV Virion Preparation

The EBV-positive B95.8 monkey B cell line was treated with 500 ng/mL of 12-O-tetradecanoylphorbol 13-acetate (TPA, Sigma, Burlington, MA, USA) for 3 h, followed by washing with Dulbecco’s phosphate-buffered saline (DPBS, Gibco, Jenks, OK) three times. The cells were then resuspended in fresh complete RPMI1640 media and cultured for 5 days before the medium was collected.

### 2.4. Primary EBV Infection

PBMCs in 12-well plates (3~4 × 10^6^ cells each well) were incubated with approximate 25 µL of EBV virions per well in the presence of 200 ng/mL of cyclosporin A (CSA, Sigma, Saint Louis, MO, USA) for up to 30 days. PBMCs with or without CD4+ depletion were seeded in 24-well plates (about 1.5 × 10^6^ cells each well) and infected with 20 µL of EBV virions per well in the absence of CSA for up to 30 days. The cells were collected at different time points for assays.

### 2.5. Co-Culture Assay

JiJoye-shCtrl or JiJoye-shIRF4 cells were pretreated with 1 µg/mL of doxycycline (DOX, ThermoFisher Scientific, Logan, UT, USA) for 3 days prior to the co-culture assay. CD4+ T cells isolated using a CD4 T Cell Isolation Kit from PBMCs of HS, HIV-IR and HIV-INR were seeded in 12-well plates (about 2.5 × 10^6^ cells each well) and co-cultured with 5 × 10^5^ pretreated JiJoye-shCtrl or JiJoye-shIRF4 cells in complete RPMI media containing 1 µg/mL of DOX, or the supernatant from these cells for 3 days, and then the cells were collected for flow cytometer analysis.

5 × 10^5^ JiJoye-shCtrl or JiJoye-shIRF4 cells were pretreated with 1 µg/mL of DOX for 2 days prior to the co-culture assay and seeded in the bottom wells of Corning 24 mm Transwells with a 0.4 µm pore polycarbonate membrane insert (Corning Incorporated, Corning, NY, USA) in the presence of 1 µg/mL of DOX. CD4+ T cells isolated from HS PBMCs were seeded in the inserts of the Transwells (about 7 × 10^6^ cells per insert). After 3 days of co-culture, the cells from both the bottom wells and top inserts were collected for immunoblotting.

### 2.6. Flow Cytometry

The harvested cells were washed with DPBS and stained with the fixable viability dye eFluor™ 450 (eBioScience, Carlsbad, CA, USA) for 30 min at 4 °C and then washed twice with DPBS.

For flow cytometry analysis, the following fluorescence-conjugated antibodies were used for cell-surface staining: PE anti-human CD19, PerCP/Cy5.5 anti-human CD4, FITC anti-human CD4, FITC anti-human PD-L1, APC anti-human PD1, FITC anti-human CD38, Alexa Fluor 700 anti-human CD38, FITC anti-human CD69, PE anti-human CD25, PerCP/Cy5.5 anti-human CD3, Alexa Fluor 700 anti-human CD8, FITC anti-human CD14, PE anti-human CD33, PerCP anti-human CD15, and APC anti-human HLA-DR. APC anti-human Ki67 and PE-Cy7 anti-human IRF4 antibodies were used for intracellular staining (all antibodies from BioLegend; San Diego, CA, USA). The cell surface staining was performed, firstly, with antibodies in Cell Staining Buffer (BioLegend) for 20 min at room temperature. Then, the cells were washed with Cell Staining Buffer and incubated with eBioScience^TM^ fixation/permeabilization buffer (Invitrogen, Carlsbad, CA, USA) for 45 min at RT. The cells were then washed with 1X permeabilization buffer (Invitrogen), and the intracellular staining was performed by incubating the cells with APC anti-human Ki67 and/or PE-Cy7 anti-human IRF4 antibodies diluted in 1X permeation buffer for 1 h at RT. The cells were washed twice with Cell Staining Buffer and then analyzed using a BD FACSymphony™ A3 flow cytometer (BD Biosciences, San Jose, CA, USA). The data were processed and analyzed using the FlowJo v10.0 software. Unstained, isotype, and single positive controls were used for gating and compensation.

### 2.7. Immunoblotting

Cells were harvested for immunoblotting (IB) with primary antibodies including rabbit anti-PD-L1, rabbit anti-PD1, rabbit anti-cGAS, HRP (horseradish peroxidase)-conjugated rabbit anti-β-actin (Cell Signaling Technology, Danvers, MA, USA), rabbit anti-STING (Abclonal, Woburn, MA, USA), rabbit anti-IRF4 (ProteinTech, Rosemont, IL, USA), and mouse anti-GAPDH (Santa Cruz Technologies, Santa Cruz, CA, USA) antibodies. After incubation with the appropriate HRP-conjugated secondary antibodies (Cell Signaling Technology, Danvers, MA, USA), protein detection was performed using the Amersham ECL Detection Reagent (GE Healthcare Biosciences, Pittsburgh, PA, USA). The protein bands were visualized with the ChemiDoc MP Imaging System (Bio-Rad, Hercules, CA, USA).

### 2.8. RNA Extraction and Real-Time Quantitative PCR

Total RNA extraction, reverse transcription, and real-time quantitative PCR (RT-qPCR) were performed, and the RT-qPCR results were analyzed, as detailed in our previous publication [65], with the following primers. LMP1: F, 5′-ACGGACAGGCATTGTTCC-3′; R, 5′-TGAGCAGGATGAGGTCTAGG-3′. IRF4: F, 5′-GAGCCAAGCATAAGGTCTGC-3′; R, 5′-AGGACCTGGTCCAGGTTGC-3′. PD1: F, 5′-GCCACCTTCACCTGCAGC-3′; R, 5′-GAGATGGCCCCACAGAGGTAG-3′. PD-L1: F, 5′-TGCCGACTACAAGCGAATTACTG-3′; R, 5′-CTGCTTGTCCAGATGACTTCGG-3′.

### 2.9. RNA Array

Total RNA was isolated using a Qiagen RNeasy Mini Kit (Qiagen, Germantown, MD, USA) and subjected to RNA array analysis by ArrayStar Inc. on a chip with a total of 40,173 lncRNA probes and 20,730 mRNA probes. For mRNAs, raw signal intensities were normalized by the quantile normalization method in GeneSpring GX v12.1. Low-intensity mRNAs were filtered. mRNAs where at least 1 out of 2 duplicate samples had flags in present or marginal were chosen for further analysis. Differentially expressed mRNAs with statistical significance that passed filtering log2(|FC| > 1.5) and −log10(*p*) > 0.5 were used for volcano plotting and GO (gene ontology) enrichment analyses.

### 2.10. Statistics

The data were analyzed using Prism 9.3 software (GraphPad, San Diego, CA, USA) and are presented as mean ± standard deviation (SD). Comparisons between two groups were conducted by paired or unpaired *t*-test, and comparisons between multiple groups were conducted by one-way ANOVA after excluding outliers that were identified by the ROUT method (Q = 1.000%). The correlation between two factors (e.g., PD-L1 and IRF4) was made by simple linear regression. Statistical significance was reported with * *p* < 0.05; ** *p* < 0.01; *** *p* < 0.001; and **** *p* < 0.0001.

## 3. Results

### 3.1. IRF4 Transcriptionally Regulates Both PD-L1 and PD1

In recent years, we have employed multiple omics approaches, including microarray, ChIP-seq, phosphoproteomics, and secondary analyses of high-throughput data, to identify IRF4 transcriptional targets and elucidate its functions in hematological malignancies and other biological settings [61]. Our RNA array results indicate that IRF4-regulated genes (by IRF4 shRNA compared with control shRNA) in EBV lymphoma cells are involved in immune response and inflammation, cell development and proliferation, antigen presentation, T cell receptor (TCR) signaling, cell cycle, and unfolded protein response (UPR) (Figure 1A,B).

Interestingly, multiple genes encoding cell surface immune checkpoint receptors and their ligands were transcriptionally regulated by IRF4, including PD-L1 (CD274), PD1 (CD279), CTLA4 (CD152), B7-2 (CD86), LAG3 (CD223), and PD-L2 (CD273), in these virus-infected cancer cells (Figure 1A). We further validated by immunoblotting (IB) that IRF4 depletion in JiJoye cells downregulated the PD-L1 and PD1 protein levels, but transient expression of IRF4 in BJAB cells upregulated their levels (Figure 1C). Interestingly, substantial PD1 levels were detected in B lymphoma cell lines (e.g., JiJoye), correlated with the IRF4 levels. Additionally, our IB results show that IRF4 depletion upregulated cGAS and the longer isoform of STING protein levels (Figure 1C). Since the cGAS-STING pathway is pivotal in cancer immune surveillance, IRF4 may also play a role in the evasion of the cGAS-STING-mediated innate immune response in EBV latency, at least by regulating their expression.

Together, these results suggest that IRF4 plays a role in subverting multiple immune surveillance mechanisms by the transcriptional control of related genes, including PD-L1 and PD1, and cGAS and STING, in the context of oncogenic viral infection that is associated with blood cancers.

### 3.2. Primary EBV Infection of PBMCs Upregulates IRF4 Levels in Both B and T Lymphocytes

To evaluate the role of IRF4 in the regulation of PD-L1 and PD1 during EBV infection, we initially examined IRF4 deregulation during EBV incubation with PBMCs isolated from healthy subjects (HS).

The flow cytometry results show that IRF4 positivity was detected in 100% of CD19+ B and CD4+ T lymphocytes before EBV infection, and its expression levels were significantly increased in both cell types as early as one day after EBV infection. The IRF4 level remained relatively high in CD19+ B cells during the establishment of lymphoblastoid cell lines (LCLs) but gradually declined in CD4+ T cells and remained at a low level until CD4+ T cells disappeared approximately two weeks post-infection (Figure 2A,B).

Ki-67, a well-established proliferation marker, was induced, and its positivity percentage peaked on day 4 in a subset of both CD19+ B cells and CD4+ T lymphocytes. In contrast to the frequency of Ki-67-positive CD19+ B cells that remained significantly elevated, the frequency of Ki-67-positive CD4+ T cells rapidly declined after a week (Figure 2A,B), and all CD4+ T cells disappeared approximately two weeks post-infection.

These results indicate that EBV infection of PBMCs in vitro induced IRF4 expression in both CD19+ B and CD4+ T lymphocytes and promoted the proliferation of both cell populations at the early stage.

### 3.3. Primary EBV Infection Impairs CD4+ T Cell Functions

We then measured the populations of PD-L1, PD1, and CD38-positive cells in CD19+ B and CD4+ T cells by flow cytometry during EBV infection of PBMCs.

The results show that EBV infection significantly increased the frequencies of the PD-L1-positive cell population in CD19+ B cells by day 7 and the PD1-positive cell population in CD4+ T cells as early as one day post-infection, and the frequencies of both populations remained at high levels throughout the first week. The frequency of the PD-L1-positive CD19+ B cell population remained at a high level in the established LCLs, and the frequency of the PD1-positive CD4+ T cell population also remained relatively stable until these cells disappeared approximately two weeks post-infection (Figure 3A,B).

CD38, the most reliable surrogate marker for T cell immune activation [66], was significantly upregulated at the early stage of infection (day 1) in a subset of both CD19+ B cells and CD4+ T lymphocytes. While the frequency of the CD38+CD19+ B cell population declined rapidly after the first day, the frequency of the CD38+CD4+ T cell population remained relatively stable throughout the first week post-infection (Figure 3A,B).

These results indicate that EBV induced B/T cell activation at the early stage but impairedCD4+ T cell functions by activating the PD-L1/PD1 immune suppression pathway during in vitro infection.

### 3.4. Depletion of CD4+ T Cells Promotes EBV Transformation of PBMCs

Since EBV infection activated the PD-L1/PD1 immune suppression pathway, this suggests that CD4+ T cells may exert a suppressive role during EBV infection and transformation. Thus, we next sought to examine whether the depletion of CD4+ T cells would enhance EBV-mediated transformation.

To test this, EBV was incubated with HS PBMCs or with CD4+ T cell-depleted HS PBMCs, followed by measuring the frequencies of cell populations expressing PD-L1, CD38, Ki-67, and IRF4, as well as B cell transformation at different time points (Figure 4). By day 17 post-infection, the CD19+ B cell population, along with the assayed markers, including CD38, PD-L1, Ki-67, and IRF4 within this population, was significantly higher in PBMCs depleted of CD4+ T cells compared with intact PBMCs (Figure 4A,B).

Consistently, the RT-qPCR results show that LMP1, IRF4, and PD-L1 were all significantly higher in PBMCs without CD4+ T cells 17 days post-infection (Figure 4C), and proliferating colonies were significantly more grown from PBMCs depleted of CD4 T cells compared with intact PBMCs by day 26 post-infection (Figure 4D).

Taken together, these results support the role of CD4+ T cells in inhibiting the in vitro EBV transformation of PBMCs.

### 3.5. EBV Lymphoma Cells Impair PBMC Immune Functions via IRF4 in Co-Culture

We next investigated whether EBV lymphoma cells can impair the immune function of PBMCs, including CD4+ T cells, through the upregulation of IRF4. To this end, we used the JiJoye cell lines stably expressing the control shRNA (JiJoye-shCtrl) and IRF4 shRNA (JiJoye-shIRF4), which we generated in our previous study [60], as the model system for co-culture assays.

We co-cultured primary CD4+ T cells with doxycycline (DOX)-pretreated JiJoye-shCtrl vs. JiJoye-shIRF4 stable cells, followed by evaluating CD4+ T cell functions after 3 days by flow cytometry and/or IB. The flow cytometry results show that the frequencies of PD-L1+, Ki-67+, and CD69+ JiJoye cell populations were all significantly lower in JiJoye-shIRF4 cells compared with JiJoye-shCtrl cells (Figure 5A). The IB results show that IRF4 depletion in JiJoye cells downregulated both PD1 and PD-L1 in both JiJoye cells and the co-cultured CD4+ T cells (Figure 5B). In addition, the frequencies of CD4+ T cell populations expressing CD69+ or CD25+ (early activation markers) were all significantly higher in co-culture with either JiJoye-shCtrl or JiJoye-shIRF4 compared with CD4+ T cells alone. Additionally, the frequencies of CD69+ and CD25+ in CD4+ T cell populations were significantly lower in co-culture with JiJoye-shIRF4 compared with JiJoye-shCtrl (Figure 5C).

We also seeded PBMCs from HS in the culture media derived from DOX-treated JiJoye-shCtrl or JiJoye-shIRF4 cells. The flow cytometry results show that the frequencies of CD14+CD33+ myeloid, early-progenitor myeloid-derived suppressive cells (e-MDSCs) that play a crucial role in immune suppression, particularly in cancer, chronic inflammation, and viral infections [67]), and the frequencies of CD3+CD8+ T cell populations were all significantly lower in co-culture with JiJoye-shIRF4 compared with JiJoye-shCtrl (Figure 5D).

These results indicate that EBV lymphoma cells impaired PBMC immune functions in co-culture by impairing CD4+ T cell functions and by upregulating immunosuppressive cell populations, and that the high level of IRF4 in these EBV lymphoma cells contributed to this effect.

### 3.6. EBV Transforms HIV PBMCs with Greater Efficiency Compared with HS PBMCs

The frequency of the CD3+CD4+ T cell population was significantly lower in PBMCs from HIV-IR patients (CD4+ T cell counts > 450 cells/mm^3^) compared with HS and was even further reduced in HIV-INR patients (CD4+ T cell counts < 450 cells/mm^3^) (Figure 6A) in contrast to the frequency of the CD3+CD8+ T cell population (Figure 6B). Meanwhile, the frequencies of the CD19+ B cell populations showed no significant differences between each other (Figure 6C). Correspondingly, the PD1 level in the CD4+ T cell population was significantly higher in HIV-IR compared with HS and further significantly increased in HIV-INR patients (Figure 6D), whereas there were no significant differences in the CD8+ T cell population (Figure 6E). The PD-L1 level in the CD19+ B cell population was significantly higher in HIV-IR compared with HS whereas there was no significant difference compared with HIV-INR (Figure 6F). In line with the claim that aberrant CD4+ T cell homeostasis is a major feature of HIV infection [68], these results indicate that the CD4+ T cell population, rather than the CD8+ T cell population, plays the key role in HIV/AIDS immune impairment, in association with the elevated PD1 levels. Surprisingly, the IRF4-positive population in CD19+ B cells from HS, HIV-IR, and HIV-INR patients had no significant differences, possibly due to the small sample sizes (Figure 6G), whereas IRF4 and PD-L1 expression levels were correlated in CD19+ B cells in PBMCs from these three groups (Figure 6H).

Next, to evaluate the effect of HIV infection on EBV transformation, we incubated PBMCs from HS vs. HIV patients (mix of IR and INR patients) with the EBV strain B95.8. The results show that EBV induced significantly more colonies and greater proliferation (Ki-67) of B cells from HIV patients (Figure 6I,J). These results indicate that the impaired immune system caused by HIV infection sets a predisposing condition, promoting EBV infection and transformation.

## 4. Discussion

The IRF transcription factor family plays crucial roles in regulating the type I interferon (IFN)-mediated innate immune response. Recent research has disclosed the intimate interaction of the immune system with various inflammation-related diseases (including cancers), which holds exciting promise for immunotherapy. The lymphocyte-specific IRF4 plays a pivotal role in immune evasion, particularly in the context of cancer, and has garnered significant attention in recent years.

The direct 2D co-culture system provides a traditional strategy for the in vitro study of the TME, which allows direct cell–cell interactions mimicking the TME. Using this system, this study has disclosed novel mechanisms responsible for the complicated regulation of T cell immune functions by IRF4 and provided novel insights into the interaction of IRF4 with EBV in the pathogenesis of lymphocyte-related diseases.

In the TME, IRF4 has been shown to affect antigen presentation and the infiltration and function of immune cells, consequently promoting or suppressing tumor progression in a context-dependent manner [52]. For example, elevated IRF4 expression has been linked to an immunosuppressive TME that favors follicular lymphoma progression [69], whereas diminished IRF4 levels impair immune surveillance and facilitate tumor progression in chronic lymphocytic leukemia [70]. In our setting, we have shown in this study that IRF4 upregulates both PD-L1 in EBV lymphoma cells and PD1 in co-cultured CD4+ T cells, in agreement with its oncogenic role in EBV+ lymphomas.

A noteworthy observation in this study was that IRF4 expression was induced in CD4+ T cells co-cultured with EBV-infected B cells, even though EBV does not infect CD4+ T cells (Figure 2B). Activated TCR is known to potently induce IRF4 expression [71]. In line with this, our results show the upregulation of the activation markers CD25 and CD69 (Figure 5A,C) and the proliferation marker Ki-67 (Figure 2 and Figure 5B), indicating the activation of CD4+ T cells in co-culture with EBV+ B lymphoma cells. Regarding the underlying mechanism(s), LMP1 is known to induce MHC-I/II expression in mice and human cells [50,57], and we have shown that LMP1 strongly induces the expression of and activates IRF4 in EBV+ B cells [60,63,64]. IRF4 is known to induce the expression of and present MHC-I/II [72], the antigens that are recognized by T cells to activate TCR signaling, which strongly induces IRF4 expression in T cells [71]. Thus, it is plausible that IRF4 mediates LMP1 induction of MHC-I/II, which further activates TCR to induce IRF4, consequently inducing PD1, in co-cultured CD4+ T cells. Another possibility is that tumor cells are known to secrete extracellular microvesicles (e.g., exosomes), which transport substances (e.g., tumor-derived dsDNA and the cGAS product cGAMP) [73,74] from the tumor cells and present MHC antigens to the other surrounding cells [75,76]. Thus, exosomes secreted by EBV lymphoma cells may deliver LMP1 and present MHC antigens to the co-cultured T cells, further activating TCR that induces IRF4. In support of this possibility, our results show that the filtered medium from JiJoye cells had similar effects compared with JiJoye cells on regulating immune cells (Figure 5D).

The ability of IRF4 to regulate immune cell function and influence the TME makes it a critical factor in immune evasion. Future directions on the interaction of IRF4 with the immune system in EBV lymphomagenesis may include a systematic and in-depth analysis of type I IFN-mediated immune dysfunction, DNA damage response, autophagy, and consequent systematic immunosenescence or inflammaging [23,77,78,79,80,81], which may be conducted in LMP1 or EBV humanized mouse models. These profound mechanistic insights are expected to identify novel signaling pathways and novel molecules. For example, we recently identified a novel natural antisense long non-coding RNA (lncRNA) for IRF4, which we named IRF4-AS1 and is under further investigation. Long-term pursuits on these studies could lead to novel therapeutic strategies by targeting the IRF4 regulatory network to enhance anti-tumor immunity and improve patient outcomes.

## Figures and Tables

**Figure 1 viruses-17-00885-f001:**
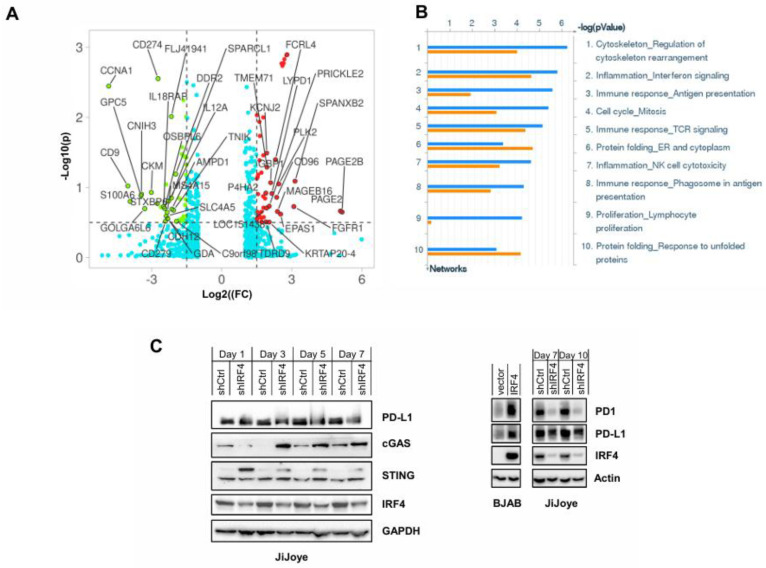
(**A**) IRF4 transcriptome in EBV-positive Burkitt’s lymphoma cell line JiJoye. Total RNAs were extracted from JiJoye-shCtrl and JiJoye-shIRF4 treated with 1 µg/mL of doxycycline for 3 days and subjected to RNA array analysis. RNA array was performed and analyzed by ArrayStar Inc. Genes with log2|FC| > 1.5 and −log10(*p*) > 0.5 were selected. (**B**) Gene ontology (GO) was analyzed for IRF4 targets, shown in the maps as blue (upregulation by IRF4) and orange (downregulation by IRF4) histograms. The height of the histogram corresponds to the relative expression value for a particular gene. (**C**) Immunoblotting validation of IRF4 regulation of PD-L1, PD1, cGAS, and the long isoform of STING in JiJoye and BJAB cells. The JiJoye-shCtrl and JiJoye-shIRF4 were treated with 1 µg/mL of doxycycline for the indicated days before cell lysates were collected and subjected to immunoblotting with the indicated primary antibodies.

**Figure 2 viruses-17-00885-f002:**
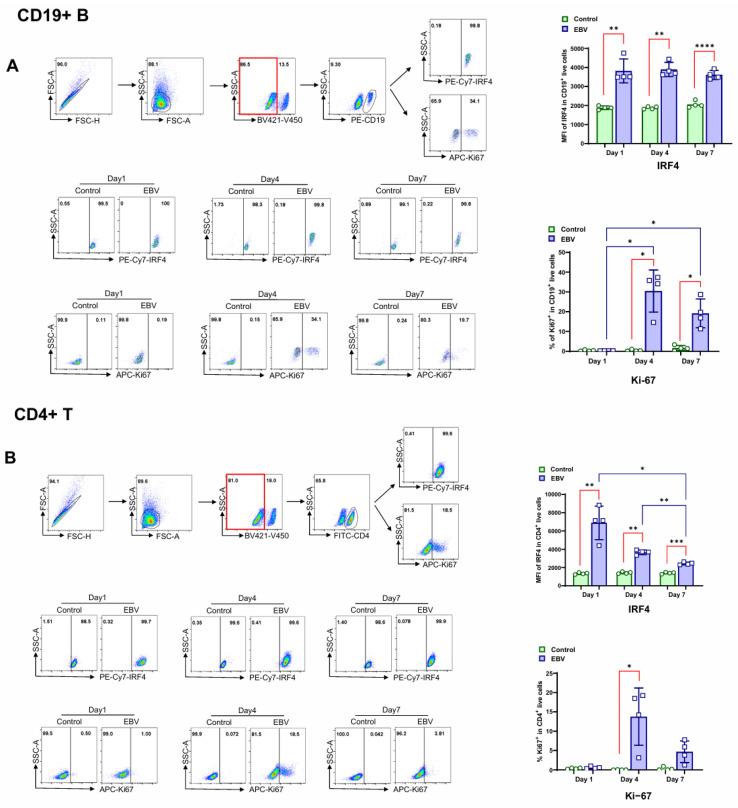
Primary EBV infection of PBMCs induces IRF4 expression in both B and T cells. PBMCs from HS (*n* = 4) were incubated with EBV for 7 days, and flow cytometry analysis was performed on day 1, day 4, and day 7. (**A**) IRF4 expression in CD19+ B cell population and Ki-67+ CD19+ B cell population were assessed by flow cytometry. (**B**) IRF4 expression in CD4+ T cell population and Ki-67+ CD4+ T cell population were assessed by flow cytometry. Data from HS were analyzed by paired *t*-test for comparison between uninfected and EBV-infected cells and by one-way ANOVA for comparisons between the samples from day 1, day 4, and day 7. * *p* < 0.05; ** *p* < 0.01; *** *p* < 0.001; and **** *p* < 0.0001.

**Figure 3 viruses-17-00885-f003:**
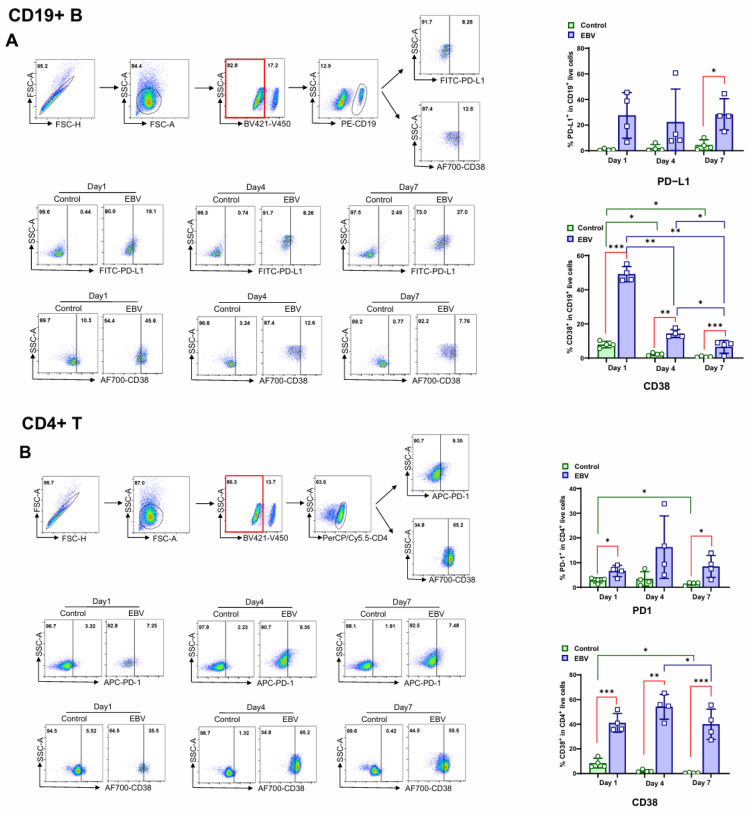
Primary EBV infection results in CD4+ T dysfunction. PBMCs from HS (*n* = 4) were incubated with EBV for 7 days, and flow cytometry analysis was performed on day 1, day 4, and day 7. (**A**) PD-L1+ and CD38+ CD19+ B cell populations were assessed by flow cytometry. (**B**) PD1+ and CD38+ CD4+ T cell populations were assessed by flow cytometry. Data from HS were analyzed by paired *t*-test for comparison between uninfected and EBV-infected cells and by one-way ANOVA for comparisons between the samples from day 1, day 4, and day 7. * *p* < 0.05; ** *p* < 0.01; *** *p* < 0.001.

**Figure 4 viruses-17-00885-f004:**
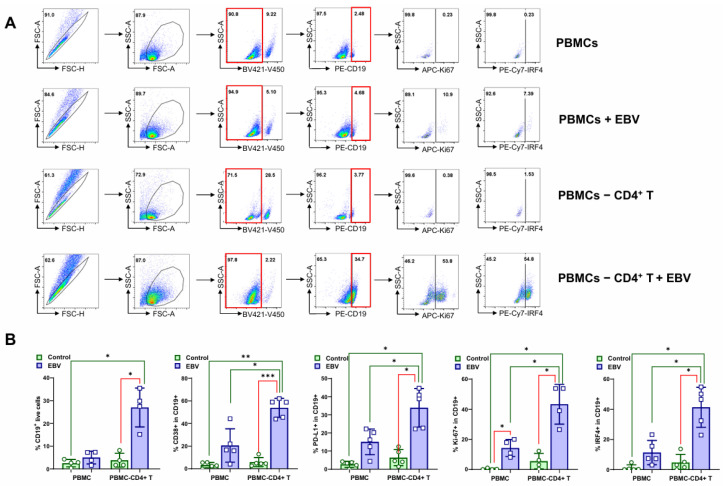
Depletion of CD4+ T cells promotes EBV transformation of PBMCs. PBMCs or CD4+ T cell-depleted PBMCs were incubated with EBV for 17 days before being subjected to flow cytometry analysis. (**A**,**B**) The CD19+ B cell population (*n* = 4) and the CD19+ B cell populations expressing CD38 (*n* = 5), PD-L1 (*n* = 5), Ki-67 (*n* = 4), or IRF4 (*n* = 5) were assessed by flow cytometry. The summarized data were analyzed by one-way ANOVA. (**C**) LMP1 (*n* = 4), IRF4 (*n* = 3), and PD-L1 (*n* = 3) expression levels in PBMCs were evaluated by qPCR. Data for LMP1 from HS (*n* = 4) were analyzed by paired *t*-test, and data for IRF4 and PD-L1 from HS (*n* = 3) were analyzed by one-way ANOVA. (**D**) EBV transformation and B cell colony formation (*n* = 5) were monitored for up to 4 weeks. Each colony number per field was the average of colonies of all fields, with 5~7 fields for each subject. * *p* < 0.05; ** *p* < 0.01; *** *p* < 0.001.

**Figure 5 viruses-17-00885-f005:**
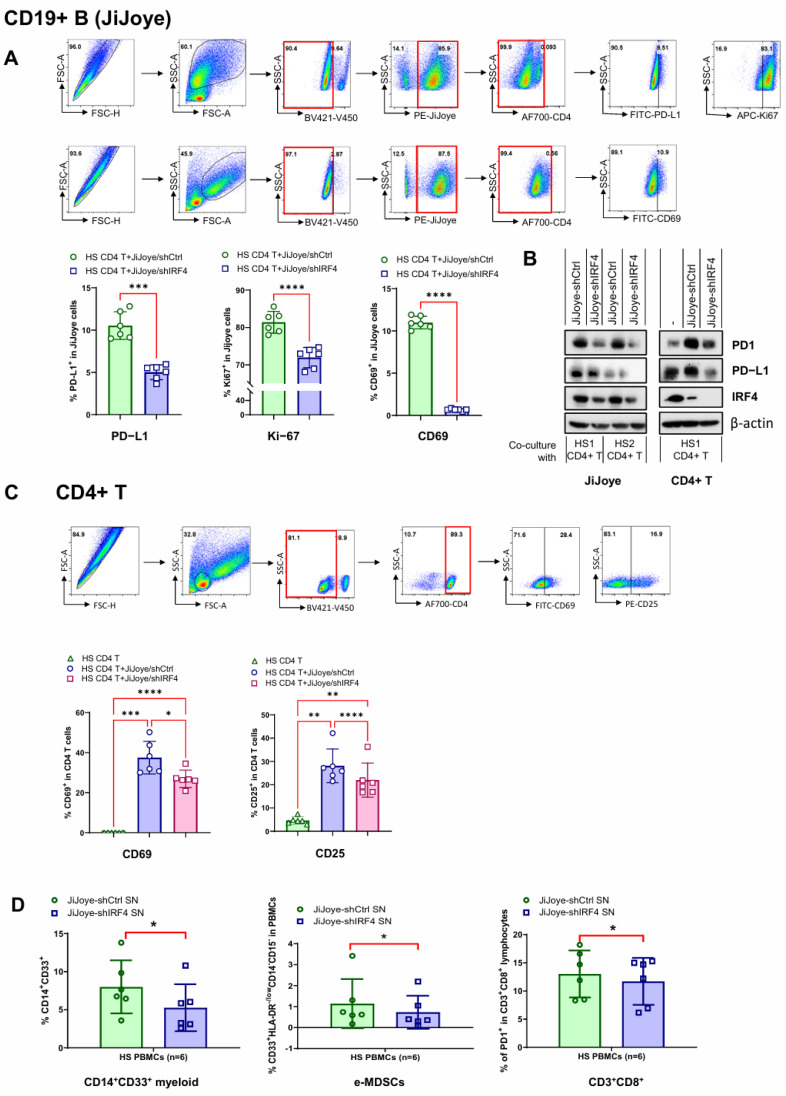
EBV lymphoma cells impair PBMC immune functions via IRF4 in co-culture. (**A**–**C**) EBV lymphoma cell line JiJoye-shCtrl and JiJoye-shIRF4 were treated with 1 µg/µL of doxycycline for 3 days, followed by co-culturing with CD4+ T cells isolated from HS (*n* = 6 for A and C; *n* = 1 for B) for 3 days. (**A**) JiJoye cell populations expressing PD-L1, the metabolic gatekeeper CD69, or the cell proliferation marker Ki-67 were assessed by flow cytometry. Data were analyzed by paired *t*-test. The representative flow images correspond to CD69. (**B**) Immunoblotting analysis for PD1 and PD-L1 protein deregulation by IRF4 in co-culture. (**C**) CD4+ T cell populations expressing the activation marker CD25 and the metabolic gatekeeper CD69 were assessed by flow cytometry. Data were analyzed by one-way ANOVA. (**D**) Cell media collected from JiJoye-shCtrl and JiJoye-shIRF4 was incubated with CD4+ T cells isolated from HS (*n* = 6) for 3 days. CD14+CD33+ myeloid, e-MDSC, and CD3+CD8+ T cell populations were assessed by flow cytometry. Data were analyzed by paired *t*-test. * *p* < 0.05; ** *p* < 0.01; *** *p* < 0.001; and **** *p* < 0.0001.

**Figure 6 viruses-17-00885-f006:**
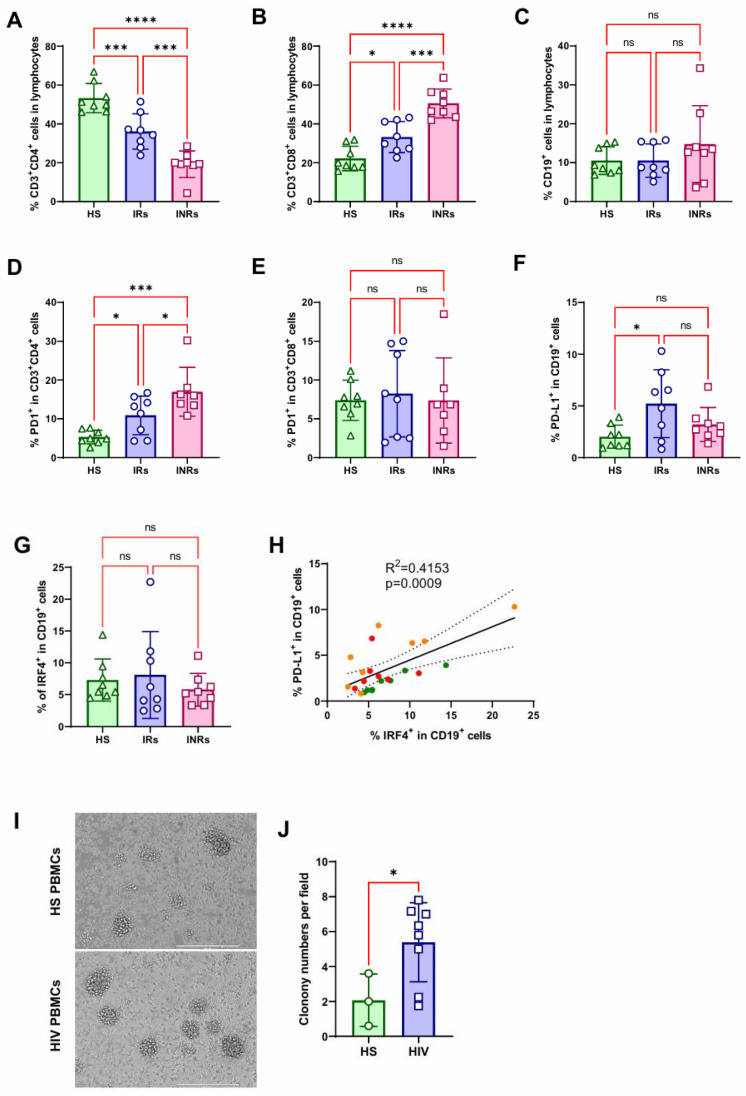
EBV transforms HIV PBMCs with greater efficiency compared with HS PBMCs. PBMCs isolated from HS, HIV-IR patients, and HIV-INR patients were incubated with EBV. (**A**–**G**) Flow cytometry was performed for the indicated cell populations. Data from HS (*n* = 8), HIV-IR (*n* = 8), and HIV-INR (*n* = 8) patients were analyzed by one-way ANOVA. (**H**) IRF4 and PD-L1 expression levels in CD19+ B cells of PBMCs from HS (green), HIV-IR (orange), and HIV-INR (red) were quantitated by RT-qPCR, and their correlation was analyzed by linear regression. (**I**,**J**) B cell transformation and colony formation were analyzed after 22 days. Each colony number per field is the average of colonies of all fields, with 5~7 fields for each subject. Data from HS (*n* = 3) and HIV (*n* = 8) patients were analyzed by unpaired *t*-test. * *p* < 0.05; *** *p* < 0.001; **** *p* < 0.0001. ns: not significant.

**Table 1 viruses-17-00885-t001:** Demographics of study subjects.

Subjects	Number	Gender	Median Age	Median CD4 T Cell Count
HS	27	15 M/12 F	45 (24–67)	N/A
HIV-IR	12	9 M/3 F	54 (40–63)	790 (650–1049)
HIV-INR	11	9 M/2 F	53 (41–74)	285 (80–433)

## Data Availability

The raw data supporting the conclusions of this article will be made available by the authors on request.

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
