# Peer review of "IRF4 Mediates Immune Evasion to Facilitate EBV Transformation"

_viruses, 2025, doi:10.3390/v17070885_

Round 1

Reviewer 1 Report

Comments and Suggestions for Authors

In the present study, the authors report interesting results regarding the impact of IRF4 expression during EBV B lymphocytes transformation. The authors initially made use of a previously established EBV+ lymphoma cell line, Jijoye, conditionally expressing an IRF4 shRNA, to investigate the impact of IRF4 on the cell transcriptome. They found that, among other genes, IRF4 transcriptionally regulates PD-L1 and PD-1 expression in the EBV+ lymphoma cell line, as well as in co-cultured CD4+ T cells. The results of the present study suggest a role for IRF4 in immune evasion during EBV latency, via the control of PD-1 and PD-L1 expression. Congruently, the authors found that the depletion of CD4+ T cells promotes the transformation of PBMCs by EBV. This suggests that CD4+ T cells play an essential role in the control of EBV B cell transformation.

Altogether, this study provides a valuable contribution to the field by offering novel insights into the role of IRF4 in the transformation of B lymphocytes by EBV.

 - Lines 185-188: The authors wrote that their RNA array indicates that “IRF4-regulated genes are involved in immune response and inflammation, cell development and proliferation, antigen presentation, TCR signalling, cell cycle, and unfolded protein response (UPR)” (refering to Figure 1, A-B). However, the Gene Ontology analysis shown in Figure 1B, only shows “Immune response” pathways. There is nothing on the other GO mentioned in the text, such as “cell cycle” or “unfolded protein response”. Could the authors provide the analysis mentioning the other GO pathways.

- Figure 1C: For sake of clarity, the authors could add a few lines in their Figure’s legend to indicate that they are using previously established Tet-inducible Jijoye-shControl and Jijoye-shIRF4 cell lines and that the days indicated correspond to the number of days post-induction of shRNA expression.

- Figure 5A: I don’t understand what the FACS images correspond to exactly. Does “PE-Jijoye” correspond to CD19 labelling? Does the first lane of FACS images correspond to Jijoye-shCtrl and the second to Jijoye-shIRF4? And if this is the case why showing only PD-L1 and KI67 assays for the first lane and CD69 for the second lane? Could the authors clarify in the Figure or in the Figure’s legend?

- The quality of the Figures makes the text in the Figure  very difficult to read. Could the authors ameliorate this point.

Typo errors

- Line 33: Shouldn’t it be “Increases” instead of “increasing”?

- Line 285: “RT- “is missing before qPCR

- Line 383: “ment’ is missing after “agree”

- Line 397: “co-cultured” instead of “c-cultured”

Author Response

We greatly appreciate the time and efforts offered by the reviewers and their valuable comments.

Reviewer #1:

 - Lines 185-188: The authors wrote that their RNA array indicates that “IRF4-regulated genes are involved in immune response and inflammation, cell development and proliferation, antigen presentation, TCR signalling, cell cycle, and unfolded protein response (UPR)” (refering to Figure 1, A-B). However, the Gene Ontology analysis shown in Figure 1B, only shows “Immune response” pathways. There is nothing on the other GO mentioned in the text, such as “cell cycle” or “unfolded protein response”. Could the authors provide the analysis mentioning the other GO pathways.

Response: Fig 1B has been replaced. Sorry for this oversight.

- Figure 1C: For sake of clarity, the authors could add a few lines in their Figure’s legend to indicate that they are using previously established Tet-inducible Jijoye-shControl and Jijoye-shIRF4 cell lines and that the days indicated correspond to the number of days post-induction of shRNA expression.

Response: Fixed.

- Figure 5A: I don’t understand what the FACS images correspond to exactly. Does “PE-Jijoye” correspond to CD19 labelling? Does the first lane of FACS images correspond to Jijoye-shCtrl and the second to Jijoye-shIRF4? And if this is the case why showing only PD-L1 and KI67 assays for the first lane and CD69 for the second lane? Could the authors clarify in the Figure or in the Figure’s legend?

Response: Sorry for the confusion. The representative flow images correspond to CD69 as marked. We add this in the legend.

- The quality of the Figures makes the text in the Figure  very difficult to read. Could the authors ameliorate this point.

Response: We included high resolution images for all figures in the submission, but they were not sent to reviewers per the journal’s procedure.

Typo errors

- Line 33: Shouldn’t it be “Increases” instead of “increasing”?

- Line 285: “RT- “is missing before qPCR

- Line 383: “ment’ is missing after “agree”

- Line 397: “co-cultured” instead of “c-cultured”

Response: All fixed. We also added a section for RT-qPCR in “M&M”. Thanks for these detailed catches.

Reviewer #2

Mechanistic Insight into IRF4 Downregulation in T cells:

Response: We understand the reviewer’s concern. We are actually developing a new technique combining nanoparticles with exosome delivery and a LMP1 mouse model to verify LMP1-IRF4-MHC and LMP1 transfer from B to T cells in our long-term plan. Given that the editor’s decision is “Minor revision”, which was requested to be returned in 5 days, we cannot include these experiments in this manuscript. However, we clarified in the revision that: (1) we did present some data supporting the activation of CD4 T cells co-cultured with EBV+ lymphoma cells, including the activation markers CD25 and CD69 (Fig 5, A ,C) and the proliferation marker Ki-67 (Fig 2; Fig 5B); (2) the two possible mechanisms for TCR activation was based on different lines of published evidence (cited in the manuscript).

Confounding Factors in HIV PBMC Transformation Assay: The increased EBV transformation efficiency in HIV-infected PBMCs is an intriguing finding. Nevertheless, interpreting this result is complicated by the fact that HIV predominantly infects CD4+ T cells, leading to a chronic state of immune dysfunction. It would be helpful if the authors could more clearly address how the presence of HIV infection in T cells may indirectly or directly influence EBV transformation in B cells. For instance, are there soluble factors or altered immune cell interactions involved? If possible, including control experiments using CD4+ T cells from HIV patients separately would aid in clarifying this point.

Response: Our results support the claim that HIV-induced immune suppression is the main factor that influences EBV transformation and IRF4 plays a role in this process (the last section in “Results”). Regarding if any soluble factors involved in this process, please see the first response above. We did include experiments from isolated CD4+ T cells (Fig 5) and from CD4+ T-depleted PBMCs (Fig 4), both (Gain-of-function and loss-of-function) supporting the importance of CD4+ T cells in this process.

Recommendation for Additional Experiments:

Whether LMP1 or other EBV components are detectable in T cells following infection or co-culture.

Whether inhibition of exosome release from EBV-infected B cells alters IRF4 expression in bystander T cells.

Direct comparison of IRF4 levels and activation status (e.g., phosphorylation) in B vs. T cells over the course of infection.

I encourage the authors to address these points through discussion and/or additional experiments.

Response: Please see the first response. Given that the editor’s decision is “Minor revision”, which is requested to be returned in 5 days, we cannot include these experiments in the revision. However, we’d like to include these interesting experiments in our long-term pursuits. We added necessary discussion in the revision.

Reviewer 2 Report

Comments and Suggestions for Authors

This manuscript presents novel findings on the role of IRF4 in EBV-mediated immune evasion and transformation, particularly highlighting its regulatory effects on PD1/PD-L1 expression and suppression of the cGAS-STING pathway. Importantly, the authors show that IRF4 activation paradoxically suppresses EBV-mediated transformation in vitro and that EBV transforms PBMCs from HIV-infected individuals with greater efficiency. These observations are significant and may offer new mechanistic insights into EBV pathogenesis under immunosuppressive conditions.

However, I would like to raise the following points for the authors' consideration:

  1. Mechanistic Insight into IRF4 Downregulation in T cells:

The authors report that IRF4 expression is initially upregulated in both CD19+ B cells and CD4+ T cells upon EBV infection, but subsequently decreases specifically in T cells. While the authors propose that LMP1 may be transferred from B cells to T cells, potentially via exosomes or antigen presentation, this remains speculative and is not directly supported by experimental evidence in the current manuscript. Given the central role of this observation in establishing the immunosuppressive role of IRF4, I believe additional mechanistic studies—such as the demonstration of LMP1 transfer or signaling activation in T cells—would significantly strengthen the manuscript’s conclusions.

  1. Confounding Factors in HIV PBMC Transformation Assay:

The increased EBV transformation efficiency in HIV-infected PBMCs is an intriguing finding. Nevertheless, interpreting this result is complicated by the fact that HIV predominantly infects CD4+ T cells, leading to a chronic state of immune dysfunction. It would be helpful if the authors could more clearly address how the presence of HIV infection in T cells may indirectly or directly influence EBV transformation in B cells. For instance, are there soluble factors or altered immune cell interactions involved? If possible, including control experiments using CD4+ T cells from HIV patients separately would aid in clarifying this point.

  1. Recommendation for Additional Experiments:

Given the importance of the IRF4 pathway in both B and T cells and its contrasting effects, I recommend that the authors consider additional experiments to explore:

  • Whether LMP1 or other EBV components are detectable in T cells following infection or co-culture.
  • Whether inhibition of exosome release from EBV-infected B cells alters IRF4 expression in bystander T cells.
  • Direct comparison of IRF4 levels and activation status (e.g., phosphorylation) in B vs. T cells over the course of infection.

These additional experiments would help substantiate the proposed intercellular communication mechanisms and solidify the conclusions regarding IRF4's immunosuppressive effects.

In summary, the study contains valuable and novel findings, but would benefit from further mechanistic clarification regarding IRF4 regulation in T cells and interpretation of transformation data in HIV-infected PBMCs. I encourage the authors to address these points through discussion and/or additional experiments.

Author Response

(The authors gave the same response as above.)
